# Design, Synthesis, and Antitumor Activity Evaluation of Proteolysis-Targeting Chimeras as Degraders of Extracellular Signal-Regulated Kinases 1/2

**DOI:** 10.3390/ijms242216290

**Published:** 2023-11-14

**Authors:** Pengming Pan, Yichao He, Tongtong Geng, Zhongtang Li, Zhongjun Li, Xiangbao Meng

**Affiliations:** State Key Laboratory of Natural and Biomimetic Drugs, School of Pharmaceutical Sciences, Peking University, Beijing 100191, China; 2011110122@pku.edu.cn (P.P.); 1910307221@pku.edu.cn (Y.H.); gtt940505@163.com (T.G.); lizhongtang@bjmu.edu.cn (Z.L.); zjli@bjmu.edu.cn (Z.L.)

**Keywords:** ERK1/2, PROTAC, antitumor, MAPK pathway

## Abstract

Inhibition of the extracellular signal-regulated kinases 1/2 (ERK1/2) alone or in combination with other targets has emerged as a promising treatment strategy for a variety of human tumors. In addition to the development of inhibitors, the development of ERK1/2 degraders is an alternative approach to decrease its activity. We synthesized proteolysis-targeting chimeras (PROTACs) as effective ERK1/2 degraders, among which **B1-10J** showed high degradative activity, with DC_50_ of 102 nM and cytotoxic IC_50_ of 2.2 μM against HCT116 cells. Moreover, **B1-10J** dose-dependently inhibited tumor cell migration. Xenograft experiments in nude mice demonstrated that **B1-10J** inhibited HCT116 tumor cell growth and achieved significant regression of tumors at a daily dose of 25 mg/kg.

## 1. Introduction

The mitogen-activated protein kinase (MAPK) pathway is one of the most essential signal transduction pathways in eukaryotic cells, capable of transducing extracellular signals and influencing a variety of fundamental cellular behaviors [1,2,3]. Extracellular signal-regulated kinase 1 (ERK1) and ERK2 are well-studied classic MAPKs. Human ERK1 is composed of 379 amino acid residues, while ERK2 is composed of 360 amino acid residues. ERK1 and ERK2 are highly homologous, sharing 84% of their amino acid sequence, and conduct mostly the same biological functions [4]. Through the rat sarcoma (RAS)–rapidly accelerated fibrosarcoma (RAF)–mitogen-activated protein kinase kinase (MEK)–ERK signaling pathway, ERK1/2 primarily regulates cell proliferation, survival, growth, metabolism, motility, differentiation, and development (Figure 1). Additionally, this pathway is associated with the formation of tumors. It is frequently abnormally activated in numerous human tumors, and mutations in RAS or RAF are common. Upon mutation, some initial cancer treatment strategies may fail [5,6,7]. As a downstream protein of RAS-RAF-MEK-ERK, ERK1/2 also plays a crucial role in the signaling network of tumor development. Compared to RAS, RAF, and MEK, ERK1/2 protein mutations occur less frequently. Increasing evidence suggests that ERK1/2 is an essential target for the development of antitumor drugs [8,9,10,11]. ERK1/2 can be targeted to treat cancer either alone or in conjunction with its upstream proteins.

Numerous ERK1/2 inhibitors are currently in development. BVD-523 (ulixertinib) is an example of orally effective, potent, and specific inhibitors of ERK1 and ERK2, with Ki values of <0.3 nM and 0.04 nM, respectively. Ulixertinib has demonstrated anticancer activity in BRAF mutant and RAS mutant cell lines in preclinical studies, as well as resistance and safety against multiple solid tumors in multiple clinical trials [12,13]. GDC-0994, an additional drug undergoing clinical trials, has an IC_50_ of 6.1 nM against ERK1 and 3.1 nM against ERK2. GDC-0994 demonstrates potent antitumor activity in HCT116 mouse xenografts and excellent PK/PD and tolerance properties [14]. Additionally, Ward et al. discovered compound 35 as a potent and selective inhibitor of ERK1/2, with an IC_50_ of <0.3 nM against ERK2. Compound 35 was investigated in a mouse antitumor study and demonstrated significant tumor regression at a daily dose of 50 mg/kg [15].

In addition to the development of ERK1/2 inhibitors, the exploration of ERK1/2 degraders is another promising strategy to modulate ERK1/2 activity. Degraders have many advantages over inhibitors, including catalytic specificity and efficiency, minimal toxicity, and the ability to overcome target accumulation and mutation. PROTACs are the most widely used techniques for the selective degradation of proteins. A bifunctional PROTAC molecule with two covalently linked ligands recruits the target protein and E3 ubiquitin ligase to induce proteasomal degradation of the target protein via the ubiquitin–proteasome system. PROTAC has emerged as a promising targeted therapy strategy for a variety of diseases, particularly cancer [16,17,18]. Moreover, Li et al. reported the click-formed proteolysis-targeting chimera (CLIPATAC) technology, which effectively degraded ERK1/2 on the basis of a covalent inhibitor [19]. This result suggests that exogenous molecules can degrade ERK1/2. Therefore, we plan to design more effective ERK1/2 degraders utilizing the PROTAC technology for the discovery of new ERK1/2 modulators.

Based on previously reported ERK1/2 inhibitors, various E3 ligands and linkers were selected to synthesize PROTAC molecules. We found that ERK1/2 ligands, linkers, and E3 ligands were all critical for the degradative activities of the resulting conjugates. Finally, compound **B1-10J** was identified as the most promising compound after screening over 32 molecules with in vitro and in vivo evaluation of antitumor activity.

## 2. Results and Discussion

### 2.1. Design and Discovery of ERK1/2 Degraders

Three reported inhibitors, ulixertinib, GDC-0994, and compound 35 (Figure 2A), were selected as the ligand components of ERK1/2 [12,13,14,15] because they have high binding affinities with ERK1/2 and relatively low molecular weights. Importantly, their X-ray cocrystal structures reveal the portions of their structures that are located on the surface of the protein, which is essential for conjugation [20]. The isopropyl moiety in ulixertinib was replaced with a 3-aminopropanyl group to generate **A** (Figure 2A) for further extension. The 1-methyl-1H-pyrazol-5-amine moiety of GDC-0994 and compound 35 was changed to a 3-(4-aminobut-1-yn-1-yl)pyridin-4-amine group (Figure 2A) based on earlier probe design. Validation using molecular docking revealed that the modified molecules **A**, **B**, and **C** bind well in the ATP pocket of ERK1/2, and the exposed amino group can extend beyond the protein pocket (Figure 2A and Appendix A). An alkyl chain with 8 to 10 carbons was selected as the linker connected via an amide bond. Cereblon (CRBN) protein ligand, von Hippel–Lindau (VHL) protein ligand derivatives, and piperlongumine derivatives were chosen as ligands for E3 ligase, and adamantane, a hydrophobic tag, was also included [21,22,23]. The combination of the abovementioned ERK1/2 ligands, linker, and E3 ligase ligands generated 12 target molecules (Figure 2B). As depicted in Figure 1, Figure 2, Figure 3, Figure 4 and Figure 5, the synthesis steps followed previous reports [14,15,21,24].

The synthesis of piperlongumine derivatives is described in Figure 1. We used a method similar to the one reported in the literature to generate compound **7** [21]. 

The reagents and conditions were as follows: (i) (Boc)_2_O, DMAP, *t*-BuOH, r.t., 2 h; (ii) H_2_SO_4_, MeOH, reflux, 3 h; (iii) K_2_CO_3_, DMF, 60 °C, 12 h; (iv) a. 2 M LiOH (a.q.), THF, 12 h; b. 0.5 M HCl (a.q.), 2 h; (v) a. ClCOCOCl, DMF, DCM, r.t., 2 h; b. *n*-BuLi, THF, −40 °C, 4 h; and (vi) TFA, DCM, r.t., 2 h.

The synthesis of compound **A** is described in Figure 2. The substitution reaction of compound **8** with tert-butyl (3-aminopropyl)carbamate produced compound **9**. Compound **12** was prepared according to previous protocols [25]. Compounds **9** and **12** were submitted to Suzuki–Miyaura cross-coupling reaction, followed by hydrolysis, to generate compound **14**. Coupling compound **14** and (S)-2-amino-2-(3-chlorophenyl)ethan-1-ol using HATU in dimethylformamide (DMF) and removing the *t*-butyloxy carbonyl (Boc) groups with trifluoroacetic acid (TFA) afforded compound **A**. 

The reagents and conditions were as follows: (i) DIEA, DMF, 70 °C, 12 h; (ii) (Boc)_2_O, DMAP, DIEA, DCM, r.t., 2 h; (iii) Pd(dppf)Cl_2_, KOAc, dioxane, 90 °C, 4 h; (iv) Pd(PPh)_3_, K_2_CO_3_, dioxane, H_2_O, 90 °C, 8h; (v) a. 2N LiOH (a.q.), THF, 12 h; b. 0.5 N HCl (a.q.), 2 h; (vi) HATU, DIEA, DCM, r.t., 12 h; and (vii) TFA, DCM, r.t., 2 h.

The synthesis of compounds **B**, **B-N**, **B-J**, **B-P**, **B-B**, and **B-BZ** is described in Figure 3. Compound **19** was prepared according to previously reported protocols [15]. Compound **19** and corresponding amino compounds were submitted to Buchwald–Hartwig cross-coupling reaction, followed by removing the Boc groups with trifluoroacetic acid (TFA), to generate the corresponding compounds.

The reagents and conditions were as follows: (i) Pd(PPh_3_)_4_, K_3_PO_4_, dioxane, H_2_O, 90 °C, 8 h; (ii) TFA, DCM, r.t., 2 h; (iii) K_2_CO_3_, 18-crown-6, dioxane, 100 °C, 12 h; (iv) a. TFA, DCM, r.t., 2 h; b. NH_4_OH, MeOH, 50 °C, 2 h; (v) NaH, DMF, r.t., 4 h; (vi) BrettPhos Pd G3, Cs_2_CO_3_, *t*-BuOH, dioxane, 85 °C, 6 h; and (vii) TFA, DCM, r.t., 2 h.

The synthesis of compound **C** was performed according to previously reported protocols and is depicted in Figure 4 [26].

The reagents and conditions were as follows: (i) Na_2_CO_3_, Pd(dppf)Cl_2_, dioxane:H_2_O (1:1), 80 °C, 1.5 h; (ii) 2 N HCl (a.q.), reflux, 2 h; (iii) AD-mix β, *t*-BuOH:H_2_O (1:1), r.t. 18 h; (iv) TBDMSCl, imidazole, DCM, 0 °C, 1 h; (v) methanesulfonic anhydride, DIEA, DCM, 0 °C, 1 h; (vi) 1 M KHMDS in THF, THF:DMF (4:1), 75 °C, 20 h; (vii) *m*-CPBA, DCM, 0 °C, 2 h; (viii) NH_4_OH, dioxane, r.t., 24 h; (ix) Pd(PPh_3_)_2_Cl_2_, CuI, DIEA, THF, r.t., 18 h; (x) K_2_CO_3_, XPhos, Pd_2_(dba)_3_, MeCN, 80 °C, 18 h; and (xi) TFA, DCM, r.t., 2 h.

The synthesis of PROTAC compounds is described in Figure 5. The substitution of iodide **30** and bromide **32**, or the formation of amides via HATU-mediated condensation reactions of acids (**7**, **29**, **31**, and **33**) with corresponding amines, was carried out to afford PROTAC compounds, respectively.

The reagents and conditions were as follows: (i) HATU, DIEA, DCM/MeOH, r.t., 8 h, and (ii) K_2_CO_3_, DMF, 80 °C, 6 h.

The compounds in Figure 2B were used to treat A375 cells at various concentrations. The Western blot analysis revealed that **B1** and **B2** induced downregulation of ERK1/2 protein in A375 cells, whereas other compounds did not (Figure 2C).

Therefore, compound **B** was used as a ligand for ERK1/2, and the ligands for the E3 ligase were CRBN ligand and VHL ligand, which were coupled with various lengths and types of linkers (Table 1). In total, 14 newly synthesized compounds were tested for their ability to degrade ERK1/2. These compounds were used to treat A375 cells for 48 h. Western blotting and quantitative grayscale analysis revealed that compounds with approximately 10 carbon chains had the highest activity with CRBN as the E3 ligand, while compounds with 10 to 14 carbon chains had similar degradation activity with the VHL ligand. Compounds linked with a PEG chain, to our surprise, had poor degradation activity (Figure 3A). To evaluate the ERK1/2 degradation activity of **B1-10**, **B2-10**, **B2-12**, and **B2-14**, the A375 cell line was treated with various concentrations of these compounds for 48 h, and the expression of ERK1/2 protein was quantified using Western blot with grayscale quantification. The results demonstrated that **B1-10**, **B2-12**, and **B2-14** exhibited high degradation activity, with DC_50_ values ranging from 300 nM to 400 nM and DC_max_ values exceeding 65% (Figure 3B). 

For PROTAC compounds using CRBN ligands as E3 ligands, the 10-carbon chain length was selected as the linker, while for VHL ligands, the 12-carbon chain length was selected as the linker. The degradative effects of some structural modifications were investigated. Firstly, the amine group linking the ERK1/2 ligand and linker was replaced with methylamino and piperazine groups. For CRBN ligands, the methylamino- and piperazinyl-modified compounds **B1-10J** and **B1-10P** showed slightly increased DC_50_ compared to the amide counterpart **B1-10**, whereas for VHL ligands, the methylamino-modified compound **B2-12J** lost its degradative activity, and the piperazinyl-modified compound **B2-10P** showed a significantly increased DC_50_ value. Secondly, when the alkynyl group on compound **B** was replaced with flexible sp^3^ N or C chains (**B1-10N**, **B1-10BZ**, and **B2-2N** in Table 2), the degradative activity disappeared. Thirdly, after the pyridine group on compound **B** was replaced with benzene and pyrazole (**B1-10B** and **B1-10BZ** in Table 2), no degradative activity was observed. These findings suggest that variations in ERK1/2 ligands can significantly affect the degradative activity of synthesized PROTACs, particularly the portion of ERK1/2 ligands located outside of the protein pocket.

### 2.2. Characterization of the Degradative Activity of ***B1-10J***

**B1-10J** was chosen as a representative compound for further characterization. It was first confirmed that **B1-10J** induced ERK1/2 degradation via the ubiquitination pathway. A375 cells were treated with or without **B1-10J**, proteasome inhibitor MG132, and E3 ligase inhibitor MLN4924 for 24 h [27,28]. The Western blot results indicated that ERK1/2 protein was only degraded by **B1-10J** alone (Figure 4A). In addition, A375 cells treated with **B1-10J** at a large range of concentrations exhibited a typical “hook” effect (Figure 4B) [29]. These findings verified that **B1-10J** induced ERK1/2 degradation through the ubiquitination pathway. Moreover, experiments were conducted to demonstrate the time-dependent degradation of ERK1/2 proteins. After A375 and HCT116 cell lines were treated with 1 μM **B1-10J**, ERK1/2 protein began to degrade at 6 h and nearly reached its highest level at 24 h. The degradation of ERK1/2 in HCT116 cells lasted up to 72 h, whereas in A375 cells, the protein level returned to normal after 48 h. Phosphorylated ERK1/2 plays a significant physiological role, and its variations were also monitored [4,30]. Interestingly, pERK1/2 exhibited an upward trend during the initial phase of the degrader treatment. pERK1/2 began to decrease as ERK1/2 was progressively degraded, but after 72 h, pERK1/2 returned to a high level (Figure 4C). To evaluate the ERK1/2 degradative activity of **B1-10J** in cancer cells, Calu-6, HCT116, and B16-F10 cancer cell lines were treated with different concentrations of **B1-10J** for 48 h, and the expression of ERK1/2 protein was quantified using Western blotting with grayscale quantification. The results demonstrated that **B1-10J** showed good degradative activity on HCT116 and Calu-6, with DC_50_ values of 102 nM and 85 nM, respectively, and weak degradative activity on B16-F10 murine melanoma cells (Figure 4D and Appendix A).

### 2.3. Effects of ***B1-10J*** on the Growth and Migration of Tumor Cells

As mentioned previously, ERK1/2 is closely associated with the formation of tumors. In numerous human cancer cells, the ERK1/2 pathway is abnormally activated. Initially, the growth-inhibiting effects of **B1-10J** on three cancer cell lines were assessed, as reported in previous studies of ERK1/2 inhibitors [14,15,25,31]. A375, HCT116, and Calu-6 cells were treated with indicated concentrations of **B1-10J** for 72 h. These cancer cells displayed variable degrees of growth inhibition, with IC_50_ values of 2.1 μM for HCT116, 15.9 μM for Calu-6, and >30 μM for A375. The insensitivity of A375 cells was attributed to the lower ERK1/2 degradation ability in A375 cells, compared with that of HCT116. In addition, the growth-inhibiting effects of **B1-10J**, **B1-10**, **B1-10P**, **B2-12**, and the “warhead” part (**B-J**, Figure 3) of **B1-10J** were evaluated in these cells (Figure 5A). Because the A375 cell line was less susceptible to the growth inhibition of **B1-10J**, it was selected to observe the effect on cell migration in a Transwell chamber. As shown in Figure 5B, A375 cell migration decreased markedly after 24 h. Using 33% acetic acid to wash crystal violet to measure the absorbance, it was found that the migration rate decreased by more than 50%. This result suggested that ERK1/2 degradation inhibited the migration of A375 cells. 

### 2.4. In Vivo Testing of ***B1-10J***

To investigate the in vivo antitumor activity of **B1-10J**, nude mice were subcutaneously xenografted with the HCT116 cell line. After the tumor volumes rose to ~90 mm^3^, the mice were divided into a blank control group, low-, medium-, and high-dose groups of **B1-10J**, and a positive control group treated with 5-fluorouracil (5-FU), a commonly used chemotherapeutic drug in colorectal cancer treatment [32]. The mice were administrated 5-FU via intraperitoneal injection three times per week, while all other groups were dosed daily. As shown in Figure 6A, only nude mice of the 5-FU group exhibited a trend of weight loss, whereas the **B1-10J** groups exhibited similar body weight variations as the blank control group, indicating that **B1-10J** was well-tolerated. **B1-10J** was able to inhibit tumor growth in a dose-dependent manner and was more effective at a 25 mg/kg dose than 5-FU. Up until day 19, 25 mg/kg of **B1-10J** showed approximately 50% tumor growth inhibition. The size of the tumors at day 20 is depicted in Figure 6C and Appendix A. These results suggested that **B1-10J** effectively inhibited the proliferation of tumors in vivo.

## 3. Materials and Methods

### 3.1. Chemistry 

General Methods. All reagents and solvents used were obtained from commercial suppliers (Bidepharmatech (Shanghai, China), Energy Chemical (The Woodlands, TX, USA), Alfa (Ronkonkoma, NY, USA), etc.) without further purification, except for special cases. Reactions were monitored via TLC. Thin-layer chromatography was carried out using TLC silica gel 60 F254 plates. Flash column chromatography was performed with 200–300 mesh silica gel. The NMR spectrum was recorded on a Bruker-400 NMR spectrometer, with TMS as an internal standard and chemical shifts reported in ppm (δ). Coupling constants (J) were reported in Hz. Spin multiplicities were described as s (singlet), br (broad singlet), d (doublet), t (triplet), q (quartet), and m (multiplet). Melting point was measured by using a X-5 micro melting point meter. High-resolution mass spectra (HRMS) were obtained using a Shimadzu LCMS-IT-TOF mass spectrometer. An Agilent high-performance liquid chromatography (HPLC) system with 5 μm C18 column was used for purity analysis. The gradient was 5% CH_3_CN (MeOH) to 95% CH_3_CN (MeOH) over 25 min at a flow rate of 1 mL/min and the wavelength was 254 nm. All end products were analyzed via HPLC and all were over 95% pure. Additional figures for ^1^H NMR, ^13^C NMR, HRMS and HPLC of compounds used for biological testing were included in the Appendix A.

### 3.2. Molecular Docking

The software Sybyl x2.0 was used for computer-aided drug design. After downloading the protein crystal structure (PDB: 6gdq), Sybyl was used for hydrogenation, dehydration, deligation, and repair to generate the docking file of the protein. Chemdraw and Chemdraw 3D were used to convert 2D small molecules into 3D forms, and energy minimization processing was performed to output a docking file for small molecules. The default docking conditions of Sybyl x2.0 were used for docking, the docking results were exported, and PyMol was used for image processing.

### 3.3. Cell Culture

A375, HCT116, Calu-6, and B16-F10 cells were purchased from American Type Culture Collection (ATCC) and cultured in DMEM high glucose media (SH30022.01, Cytiva, Shanghai, China) supplemented with 10% fetal bovine serum (10099-144, Gibco, Carlsbad, CA, USA) and 1% penicillin/streptomycin (Solarbio, Beijing, China) in an incubator at 37 °C with 5% CO_2_. 

### 3.4. Western Blot and Protein Degradation Assay

Cells were cultured under the indicated conditions as described in “Cell culture”. Cells were seeded in 6-well plates (3516, Corning, New York, NY, USA). Compounds were dissolved in DMSO for storage (10 mM). Following compound treatment, cells were washed with PBS, scraped from the plate, and lysed using cell lysis buffer (Applygen, Beijing, China). After the protein in the cell lysate was quantified via the BCA quantification method, the samples were placed in a metal bath at 100 °C and heated for 5 min. Cell lysates were separated using a 10% SDS–PAGE gel and transferred to PVDF membranes (Merck Millipore, Burlington, MA, USA). The PVDF membrane was blocked with 5% nonfat dry milk in TBST for one hour. The PVDF membrane was transferred to the antibody diluent and incubated overnight at 4 °C. PVDF membranes were washed three times with TBST for 5 min each and then incubated with the appropriate HRP-conjugated secondary antibody for 1 h at room temperature. After washing three more times with TBST, the PVDF membrane was stained with a hypersensitive ECL chemiluminescence reagent (Biodragon, Beijing, China) and then imaged with Tanon 5200. Grayscale quantitative analysis was performed using ImageJ software. Statistical analysis was performed using GraphPad Prism 8. DC_50_ and D_max_ values were fitted using a four-parameter [inhibitor] versus response and reported directly from the Prism output. Mean ± SD and unpaired t tests were performed in GraphPad Prism.

### 3.5. Cell Proliferation Assay

A 96-well plate (3599, Corning, New York, NY, USA) was inoculated with 100 μL of complete medium containing 10,000 cells, and the experimental, control, and blank groups were established. Three parallel wells were used per concentration. After 10 h, the media from the experimental and control groups were replaced with a compound containing the indicated concentration of the complete medium. Cells were incubated for 72 h before 10 μL of CCK8 reagent (CA1210, Solarbio, Beijing, China) was added to each well and incubated for 1 h. Using a microplate reader (Tecan, Mannendorf, Switzerland), the absorbance at 450 nm was measured. The IC_50_ values were determined using GraphPad Prism 8 and nonlinear regression curve fitting.

### 3.6. Transwell Migration Assay

A total of 100 µL of FBS-free DMEM high-glucose medium containing 20,000 cells was added to the upper layer of the Transwell chamber of a 24-well plate (3422, Corning, New York, NY, USA), and a complete medium containing 10% FBS was added to the lower layer. The experimental group and the control group were established. The control group used DMSO (DMSO concentration in the culture medium was no more than 0.5% in each well), and the specified concentration of compounds was added to the medium of the experimental group. The cells were then placed in an incubator for 24 h. The cells were washed with PBS, the cells in the upper layer of the Transwell chamber were wiped off with a cotton swab, and the cells were fixed with cold methanol for 15 min. The cells were stained with crystal violet reagent and observed and photographed under a microscope. Then, crystal violet was washed with 33% acetic acid and the absorbance was measured at a wavelength of 590 nm. Graphing with GraphPad Prism 8 after counting cells was performed.

### 3.7. Xenograft Mouse Studies

Male BALB/c nude mice of around 8 weeks old were purchased from Charles River (Beijing, China). Nude mice were raised in SPF-grade animal rooms and had access to water and food ad libitum. HCT116 colorectal cancer cells were grown in DMEM supplemented with 10% fetal calf serum and 1% penicillin/streptomycin under standard cell culture conditions. HCT116 xenografts were prepared by subcutaneously injecting 5× 10^6^ cells suspended in PBS into the right flank of each male BALB/c nude mouse. Tumor growth was monitored via caliper measurement, and volume was calculated using the following equation: 0.5 × max (length/width) × min (length/width) × min (length/width). When the tumors reached an average of approximately 100 mm^3^, the mice were randomized into groups of five. The mice were injected intraperitoneally once a day at doses of 6, 12, 25 mg/kg compound **B1-10J** and 25 mg/kg 5-FU. Body weights and tumor volumes were measured every 2–3 days. All studies were appropriately statistically powered.

### 3.8. Statistical Analysis 

Statistical analyses were performed using GraphPad Prism software 8. The significance analysis was conducted using a two-tailed unpaired *t*-test. *p* < 0.05 was considered statistically significant (* *p* < 0.05, ** *p* < 0.01, and *** *p* < 0.001).

## 4. Conclusions

Based on the previously reported ERK1/2 inhibitors, molecular docking was used to design multiple PROTACs containing various E3 ligands. Several effective ERK1/2 degraders, such as **B1-10**, **B1-10J**, **B1-10P**, **B2-10**, **B2-12**, and **B2-14**, were identified from 24 synthesized molecules, among which **B1-10J** was the best one; its ERK1/2 DC_50_ values were 102 nM and 85 nM for HCT116 and Calu-6, respectively. We confirmed that the degradation of ERK1/2 induced by **B1-10J** in cancer cells was via the ubiquitination pathway. **B1-10J** significantly prevented the migration of A375 cells at 1 μM, and inhibited the proliferation of HCT116 cells both in vitro and in vivo. In sum, **B1-10J** is a promising lead compound for the degradation of ERK1/2 and warrants further investigation.

## Data Availability

Data are contained within the article.

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
