# Peer review of "Design, Synthesis, and Antitumor Activity Evaluation of Proteolysis-Targeting Chimeras as Degraders of Extracellular Signal-Regulated Kinases 1/2"

_ijms, 2023, doi:10.3390/ijms242216290_

Round 1

Reviewer 1 Report

Comments and Suggestions for Authors

Manuscript “Design, Synthesis and Antitumor Activity Evaluation of 2 ERK1/2 PROTAC Degraders” has aimed to development of ERK1/2 degraders as a potential efficient anticancer agent. The authors were able to synthesize and to evaluate antitumor activity of several novel PROTAC agents.

The manuscript is written in a concise manner and represents synthetic efforts as well as biological testing discussions. Synthesized compounds are sufficiently characterized by modern spectral and chromatographical methods.

The authors should emphasize the novelty of the developed PROTACs – is this the first example ERK1/2 degraders?

In the paragraph 2.1 I would recommend to check substituents names according IUPAC rules – lines 81,83 etc. The term “group” or “substituent” should be added to the corresponding chemical term.

Comments on the Quality of English Language

please check the English thtrough the text. For example, 
Line 134 The synthesis of PROTACs compounds were described in Scheme 5.
should be changed to "is described" etc.

Author Response

  • The authors should emphasize the novelty of the developed PROTACs – is this the first example ERK1/2 degraders?

Response: On line 61, we described “Moreover, Li et al. reported the click formed proteolysis targeting chemicals (CLIPATACs) technology which effectively degraded ERK1/2 on the basis of a covalent inhibitor”, which may be the first degrader for ERK1/2. But this discovery aimed to validate the concept of CLIPATACs. We designed new ERK1/2 degraders from the perspective of medicinal chemistry.

  • In the paragraph 2.1 I would recommend to check substituents names according IUPAC rules – lines 81,83 etc. The term “group” or “substituent” should be added to the corresponding chemical term.

Response: Revised as suggested.

  • please check the English through the text. For example, Line 134 The synthesis of PROTACs compounds were described in Scheme 5. should be changed to "is described" etc.

Response: Revised as suggested.

Reviewer 2 Report

Comments and Suggestions for Authors

In this article Pan et. al., and colleagues have synthesized PROTACs for ERK1/2 degradation and out of many, they selected one PROTAC (B1-10J) based on the degradative activity on ERK1/2. Further, they demonstrated growth and migration inhibition effect of B1-10J on different tumor cell lines (A375, Calu-6, HCT-116).  The authors also tested the efficacy of the PROTAC in nude mice xenograft model. The study is well designed and explained.

 I would like to mention Figure 4A. - treatment of A375 cells with MG132 at the given concentration is not increasing the accumulation of ERK1/2 in comparison with the untreated control. The authors should try different concentration of MG132.

Comments on the Quality of English Language

The english looks good to me.

Author Response

Thank you for your evaluation of our work

  • I would like to mention Figure 4A. - treatment of A375 cells with MG132 at the given concentration is not increasing the accumulation of ERK1/2 in comparison with the untreated control. The authors should try different concentration of MG132.

Response: We used MG132 to inhibit proteasomes, thereby demonstrating that our compound works through the ubiquitin-proteasome system. MG132 did not affect the expression level of ERK1/2 which corresponded to our expectations. The concentration used for MG132 was based on previous references.

Reviewer 3 Report

Comments and Suggestions for Authors

The in vitro readout measuring the efficacy of the B1-10J protac uses Western blot image densitometry as a quantitative readout for the protac's degradation efficacy. See Figures 3A, 3B and Fig.4.

This is a very old and highly inaccurate way to measure protein amounts, and definitely not quantitative. If there were various known amounts of ERK1/2 protein loaded on the same gel, this would still render the method only semi-quantitative rather than quantitative. This would be evident if the same loadings were probed with different anti-ERK1/2 primary (and different secondary) antibodies: in some cases, the differences would be smaller or bigger than with other antibodies. Also, variations of the ECL exposures can greatly manipulate differences. For example a twofold increase in band intensity could mean for example only-30-40% increase or near two-fold increase of the actual protein amount in reality. All what Western blots are good for are to show if a sample has more or less amounts of the given protein compared to a chosen control, but it is not suitable to quantitatively measure the difference with a sufficient accuracy.  For a truly quantitative measurements one can use ELISA for example.

The ERK1/2 levels by the way are not reflective of the whole cell ERK1/2 levels, only that of the soluble fraction of the cell lysates. Part of ERK1/2 gets lost during sample processing being trapped in the lysate pellet consisting of the insoluble, membrane-bound or cytoskeleton-bound fractions of ERK1/2, which are in no way inert functionally. The authors should explicitly state that they are "measuring" and showing the soluble fraction of ERK1/2. 

With all that being said, the authors can get away with this if they would change the label on the Y axis of their charts in Fig. 3 and Fig. 4 from "Levels of ERK1/2" to "Densitometry levels of ERK1/2 Western blot".

Regarding the in vivo part of their study, the results plotted on the graph come from tumor collection with unavoidable variability in size, since it is difficult to control for individual differences of drug uptake from the peritoneum and of the differences in vascularization of the implanted tumors. For example, panel B of Fig. 6 shows that the mice treated with 5-FU had 2 really big tumors and 5 relatively small ones, which on average were smaller than most of the tumors from mice treated with 25 mg/kg B1-10J, indicating that the difference was skewed due to the two big tumors in the 5-FU group. Using more mice per group would provide a more accurate picture.

The authors should write a more detailed and carefully worded description of their findings in Section 2.4 and they also should add a more restrained and careful version of the "Conclusions" section.

Comments on the Quality of English Language

Only minor changes are required.

Author Response

  • With all that being said, the authors can get away with this if they would change the label on the Y axis of their charts in Fig. 3 and Fig. 4 from "Levels of ERK1/2" to "Densitometry levels of ERK1/2 Western blot".

Response: We agree with your opinion on Western blot. WB is indeed a semi quantitative method. The main reason why we adopted this method is: 1. In previous reports on PROTACs, most of the work used WB for semi quantitative analysis. 2. WB provides readers with a more intuitive display of protein changes. We have taken your suggestion and would change the label on the Y-axis from "Levels of ERK1/2" to "Densitometry levels of ERK1/2 Western blot".

  • Regarding the in vivo part of their study, the results plotted on the graph come from tumor collection with unavoidable variability in size, since it is difficult to control for individual differences of drug uptake from the peritoneum and of the differences in vascularization of the implanted tumors. For example, panel B of Fig. 6 shows that the mice treated with 5-FU had 2 really big tumors and 5 relatively small ones, which on average were smaller than most of the tumors from mice treated with 25 mg/kg B1-10J, indicating that the difference was skewed due to the two big tumors in the 5-FU group. Using more mice per group would provide a more accurate picture.

Response: We strongly agree that it is difficult to control individual differences of xenograft mouse model regardless oral, or IP administration. The tumors size and growth curve of 5-Fu group in this manuscript are consistent with those reported data. (PMID: 15720820, 36700552)

  • The authors should write a more detailed and carefully worded description of their findings in Section 2.4 and they also should add a more restrained and careful version of the "Conclusions" section.

Response: Section 2.4 was revised as suggested. A rewritten version of the "Conclusions" section was highlighted in the revised manuscript.